# Therapeutic Drug Monitoring of Antimicrobials in Critically Ill Obese Patients

**DOI:** 10.3390/antibiotics12071099

**Published:** 2023-06-24

**Authors:** Julie Gorham, Fabio S. Taccone, Maya Hites

**Affiliations:** 1Department of Intensive Care, Hôpital Universitaire de Bruxelles (H.U.B), 1070 Brussels, Belgium; julie.gorham@hubruxelles.be (J.G.); fabio.taccone@hubruxelles.be (F.S.T.); 2Clinic of Infectious Diseases, Hôpital Universitaire de Bruxelles (H.U.B), 1070 Brussels, Belgium

**Keywords:** obesity, antibiotics, antifungals, pharmacokinetics/pharmacodynamics, therapeutic drug monitoring, dosage

## Abstract

Obesity is a significant global public health concern that is associated with an elevated risk of comorbidities as well as severe postoperative and nosocomial infections. The treatment of infections in critically ill obese patients can be challenging because obesity affects the pharmacokinetics and pharmacodynamics of antibiotics, leading to an increased risk of antibiotic therapy failure and toxicity due to inappropriate dosages. Precision dosing of antibiotics using therapeutic drug monitoring may help to improve the management of this patient population. This narrative review outlines the pharmacokinetic and pharmacodynamic changes that result from obesity and provides a comprehensive critical review of the current available data on dosage adjustment of antibiotics in critically ill obese patients.

## 1. Introduction

Obesity is characterized by excessive accumulation of body fat and is typically classified using various size metrics such as body mass index (BMI), body surface area (BSA), total body weight (TBW), a percentage of ideal bodyweight (IBW), adjusted bodyweight (ABW), lean bodyweight (LBW) and predicted normal weight (PNWT) [1,2]. BMI is calculated by dividing TBW in kilograms by height in meters squared (i.e., kg/m^2^), and overweight is defined as a BMI ≥ 25 kg/m^2^, while obesity is defined as a BMI ≥ 30 kg/m^2^. According to the World Health Organization (WHO), the prevalence of obesity is increasing globally, with 13% of adults affected in 2016, and one-third in the United States [3,4]. Obesity is a significant public health issue that is responsible for increased morbidity and mortality [5]. It also increases the risk of various infections, such as bacteremia, skin, surgical site, community-related infections, and healthcare-related infections [6,7,8], which can result in ICU admission. In critically ill patients with severe infections, achieving adequate antimicrobial concentrations is vital to treatment success. However, managing infections in critically ill obese patients is challenging due to the pharmacokinetic (PK) variations induced by both critical illness and obesity on the antimicrobials. There are limited data on optimal dosing for this patient population, making it a crucial area for research. Therefore, the purpose of this review is to identify PK changes caused by obesity and provide a critical review of the available data on dosage adjustment of antimicrobials guided by therapeutic drug monitoring (TDM) in critically ill obese patients.

## 2. Methods

A literature search was conducted on PubMed/MEDLINE from inception to May 2023 in order to retrieve prospective or retrospective studies, case series/reports, and clinical trials concerning the use and dosing adjustments guided by TDM of antimicrobials in critically ill obese patients. Only articles published in English were included. No additional analyses on the risk of bias of each study or a meta-analysis of the existing data was performed, as the available literature is limited and the intention of this review was purely descriptive.

## 3. The Concept of PK/PD Changes in Critically Ill Patients

Critically ill patients may experience several PK changes that can impact antimicrobial dosing, particularly alterations in volume of distribution (Vd) and clearance (CL) [9,10,11]. Capillary leak syndrome, fluid resuscitation, mechanical ventilation, burn injuries, hypoalbuminemia and extracorporeal circuits can all increase the Vd of hydrophilic drugs, leading to a decrease in their plasma drug concentration [12,13,14,15]. The CL of a drug from the blood is dependent on the properties of the drug and involves the liver and kidneys, with hydrophilic antimicrobials primarily cleared renally, while lipophilic antimicrobials are primarily cleared hepatically [1]. Critically ill patients may experience both extremes of renal clearance, including acute renal failure (ARF) and augmented renal clearance (ARC). In the case of multiple organ dysfunction syndrome, including renal and/or hepatic dysfunction, antimicrobial clearance may be reduced, potentially leading to toxicity due to antimicrobial accumulation. Mechanical ventilation may also decrease antimicrobial clearance [12] due to its effect on positive expiratory pressure, which could lead to a decrease in glomerular filtration rate. 

The drug elimination half-life (T_1/2_) equation, i.e., T_1/2_ = 0.693 × Vd/CL, suggests that increased drug clearance is likely to reduce drug elimination half-life, whereas an increased Vd is likely to increase it. Hypoalbuminemia, which is commonly seen in critically ill patients, can affect the PK of highly protein-bound antimicrobials due to drug-protein binding. In cases of hypoalbuminemia, the free/unbound proportion of drugs increases in the serum, and only unbound drugs are able to distribute into body tissues and be eliminated from the vascular compartment [16]. Therefore, hypoalbuminemia is a factor that contributes to the greater Vd and CL of many antimicrobials [17,18] in critically ill patients compared to non-critically ill patients. Factors introducing PK/PD changes have been summarized in Figure 1.

Increases in Vd and CL are likely to decrease the maximum plasma drug concentration (C_max_), potentially leading to underdosing and therapeutic failure, particularly in critically ill patients. For time-dependent antibiotics such as β-lactams, increased clearance may lead to a reduced time, during which the concentration of the unbound drug is maintained above the minimal inhibitory concentration (MIC) of the infecting pathogen. For concentration-dependent antibiotics whose bactericidal effect depends mainly on the maximal concentration achieved, the C_max_/MIC ratio may be decreased due to the increased Vd. The AUC/MIC ratio may also be affected, as AUC is a function of CL. Attaining the AUC/MIC ratio may be particularly important in critically ill patients to prevent the development of bacterial resistance [19]. In this setting, obesity per se is another factor that can alter antimicrobial PKs.

## 4. PK/PD Changes in Obese Patients

Obesity can also contribute to PK variations in antimicrobials due to an increase in adipose and lean muscle tissue. For instance, the Vd of lipophilic antimicrobials can be increased in obesity due to an increase in adipose tissue, whereas an increase in lean mass may increase the Vd of hydrophilic antimicrobials [20]. In obese patients, the accumulation of fat in the liver can alter hepatic blood flow and slow drug metabolism. An increase in cytochrome P450 CYP2E1 and a decrease in cytochrome P450 CYP3A4 activity have also been documented, which may modify the CL of antimicrobial drugs [21]. An increase in renal clearance has been attributed to increased organ mass and renal blood flow in obese patients, but a decrease in renal clearance is sometimes observed due to chronic renal dysfunction in patients with comorbidities [22]. As a result of these changes in Vd and Cl, the T_1/2_ of antibiotics may be reduced or increased in obesity.

Studies are contradictory regarding how obesity can affect plasma binding proteins. While obesity does not seem to impact drug binding to albumin [23], some studies have described changes in protein binding due to increased plasma concentrations of α1-acid glycoprotein and free fatty acids, which can modify Vd [24,25]. However, even though both critical illness and obesity can cause PK changes, it appears that sepsis is primarily responsible for the PK variations observed in obese critically ill patients for antibiotics, such as meropenem, piperacillin, and cefepime/ceftazidime, rather than obesity itself [26,27], even in patients with a BMI ≥ 40 kg/m². PK/PD changes have been summarized in Figure 1.

In situations of extreme PK/variability, where higher than standard doses may be required [28], therapeutic drug monitoring (TDM) can provide an opportunity to achieve therapeutic antimicrobial exposure and avoid the emergence of resistance and toxicity. In a population of obese critically ill patients, higher empiric doses of vancomycin were associated with a higher risk of vancomycin-induced nephrotoxicity [29,30].

## 5. Therapeutic Drug Monitoring

Therapeutic drug monitoring of anti-infectious agents has been used since the 1960s to prevent toxicity, therapeutic failure and the emergence of bacterial resistance [31]. Despite the increasing evidence of TDM’s utility, its routine use is limited due to cost, availability, long turnaround time to obtain results in many laboratories (except for aminoglycosides and glycopeptides), and limited data showing favorable clinical outcomes when TDM is performed [32,33]. However, drugs with a narrow therapeutic range, a demonstrated correlation between plasma concentration and efficacy as well as toxicity, and an unpredictable pharmacokinetic profile (e.g., critically ill and obese patients) [34,35] are appropriate candidates for TDM. Guidelines recommend that vancomycin TDM should be performed in patients receiving concomitant nephrotoxic agents, those with burns or altered renal function, ICU and obese patients (Class 1C recommendation), and elderly patients and those with concomitant hepatic disease (Class 2C recommendation) [36]. Several other guidelines on TDM [28,36,37] for antimicrobials are also available, reporting similar recommendations.

In critically ill patients, TDM is routinely recommended when aminoglycosides, β-lactams, glycopeptides, and voriconazole are administered. For fluoroquinolones, polymyxins, antivirals, and all antifungals except voriconazole, TDM is neither recommended nor discouraged [28].

## 6. TDM in Obese Critically Ill Patients

### 6.1. Antifungal Agents

#### 6.1.1. Echinocandins

Caspofungin, micafungin, and anidulafungin have demonstrated concentration-dependent reduction of fungal growth [38], with the AUC/MIC ratio used as the PK/PD target [39]. Some studies have reported suboptimal dosing of echinocandins at the current recommended doses in specific groups, such as critically ill and obese patients, due to PK variability [40,41,42,43,44,45,46,47,48,49]. Higher loading doses (LD) and maintenance doses (MD) were necessary to achieve the PK and pharmacodynamic (PD) targets. As there is no consensus on the exact dose that should be administered to this specific population of obese critically ill patients, TDM should be considered [50,51] to determine appropriate dosages.

#### 6.1.2. Azoles

Among the azoles, only TDM of voriconazole should be routinely performed. The standard dosing regimen (SDR) for invasive fungal infections is a loading dose (LD) of 6 mg/kg BID on the first day followed by 3–4 mg/kg BID. The optimal trough concentration (C_min_ target) for TDM is between 2 and 6 mg/L. For itraconazole, isavuconazole, posaconazole, and fluconazole, there are no routine recommendations for TDM [52] even though altered PK/PD of azoles have also been demonstrated in specific groups such as critically ill patients [49,51]. For itraconazole, a C_min_ range > 0.5–1 mg/L is considered the optimal therapeutic serum concentration, and a C_min_ > 1 mg/L for posaconazole [28]. However, according to our knowledge, there are currently no available studies evaluating TDM of antifungals in a population of obese critically ill patients.

### 6.2. Antiviral Agents

To our knowledge, there are no available studies regarding dosage recommendations for antiviral agents in critically ill obese patients, nor on the use of TDM, particularly for acyclovir, foscarnet, and ganciclovir. However, considering that obesity alters the PKs of many drugs, the administration of higher doses could be considered. In such cases, TDM should be performed to optimize patient outcomes and avoid toxicity. Further studies are needed to evaluate the use of TDM for antivirals in this patient population.

### 6.3. Antibacterial Drugs

#### 6.3.1. β-Lactams

TDM-guided dosing has been associated with improved clinical and microbiological cure rates, treatment failure, and target attainment in critically ill patients receiving β-lactam antibiotics [53]. Higher doses of β-lactams or extended infusions may be necessary to treat infections caused by less susceptible pathogens in obese critically ill patients [54,55,56,57,58]. Therefore, TDM should be routinely performed in this patient population [26], especially as BMI has been identified as a factor contributing to PK/PD target non-attainment in critically ill patients [59]. However, there are few studies evaluating β-lactam TDM in this vulnerable population.

Hites et al. [26] demonstrated that the SDRs of meropenem, ceftazidime, cefepime, and piperacillin-tazobactam in obese patients resulted in underdosing and overdosing in approximately one-third and one-fourth of the patients, respectively. The authors also showed that the total daily doses of β-lactams needed to attain PK/PD targets were the same in both the obese cohort (BMI > 30 kg/m^2^) and the non-obese cohort (BMI < 25 kg/m^2^) of critically ill patients. However, meropenem serum concentrations were significantly lower in obese patients not on renal replacement therapy (RRT) when compared to non-obese patients. Conversely, Alobaid et al. [60] compared piperacillin and meropenem trough concentrations and PK/PD target achievement in critically ill obese (BMI > 30 kg/m^2^) and non-obese patients; the authors found that obese patients had lower piperacillin trough concentrations and a lower target attainment (100% fT > 4 × MIC), but no difference was found between patients receiving meropenem.

#### 6.3.2. Glycopeptides

Given that the CL and Vd of vancomycin are increased in obesity [61], it is recommended that the LD be based on TBW, with a risk of nephrotoxicity when a dose of ≥4 g/day is administered [29]. The SDR of vancomycin is a LD of 30–35 mg/kg followed by a maintenance dose (MD) based on creatinine clearance (CrCL) and TDM [62,63]. Clinical studies have shown that for intermittent infusion of vancomycin, a C_min_ > 10 mg/L or ≥15–20 mg/L for severe infections and for continuous infusion, a steady-state concentration (Css) between 20–25 mg/L achieves clinical efficacy and avoids toxicity [28]. However, there is limited literature on vancomycin TDM in obese critically ill patients [63,64,65,66,67,68].

A study comparing a revised vancomycin dosing protocol (10 mg/kg twice daily or 15 mg/kg once daily) to an original protocol (15 mg/kg twice daily or thrice daily) in 138 obese patients, including 31 critically ill patients, showed that the revised protocol resulted in higher therapeutic (59% vs. 36%, *p* = 0.006) and subtherapeutic (23% vs. 9%, *p* = 0.033) trough concentrations and lower excessive (18% vs. 55%, *p* < 0.001) trough concentrations than the original protocol [66]. A retrospective study of 48 obese patients, including 33% critically ill patients, treated with a new dosing strategy of vancomycin to avoid excessive trough concentrations in obese patients [67]––namely, a LD of 25 mg/kg based on TBW and an MD of 10 mg/kg––showed that therapeutic and subtherapeutic trough concentrations were attained for 35.4% and 56.3% of patients, respectively. The authors also demonstrated that age and renal function influence the trough concentration obtained, but not the weight or percentage of TBW above the ideal body weight. However, another study by Tafelski et al. [68] showed that obese critically ill patients achieved the target trough concentrations of vancomycin (10–20 mg/dL) less often than non-obese critically ill patients. A study of 150 obese patients (BMI ≥ 30 kg/m^2^) treated with vancomycin, including 33 critically ill patients, compared a pre-intervention group (trough dosing only) and a post-intervention group (peak and trough vancomycin serum concentrations), and demonstrated that measuring two serum vancomycin concentrations in this population of obese patients significantly improves target trough concentration attainment [69]. Therefore, TDM should be performed routinely in critically ill obese patients receiving vancomycin to optimize patient outcomes and avoid toxicity, but more studies are needed to evaluate the optimal dosing and TDM strategy in this population.

#### 6.3.3. Quinolones

In a case report evaluating quinolones in a critically ill obese patient (BMI 53.7 kg/m^2^) receiving a greater than SDR of ciprofloxacin (800 mg BID) and renal replacement therapy (RRT) for *Enterobacter aerogenes* lumbar osteomyelitis, authors recommended higher than SDR to treat this infection due to a less susceptible pathogen and to achieve the PK/PD target. Although it is difficult to know in this case whether it is obesity, CRRT, or less susceptible pathogen that makes higher doses necessary, TDM could be helpful in this population of obese patients to avoid toxicity and treatment failure [70]. Indeed, in another case report of a critically ill obese adolescent (BMI 72 kg/m^2^) treated with levofloxacin for an intra-abdominal *Pseudomonas aeruginosa* abscess, adjustment of the antibiotic regimen through TDM allowed a successful clinical outcome. A C_max_/MIC of 8–12, 30 min after the end of the infusion, was related to a better clinical cure in this setting [71].

#### 6.3.4. Aminoglycosides

Obesity can cause PK/PD variations of hydrophilic agents, such as aminoglycosides. Therefore, literature supports the need for aminoglycoside dosage modification in critically ill obese patients, but data are limited [72,73,74,75]. A loading dose of 7–8 mg/kg for tobramycin and gentamicin and 25 mg/kg for amikacin, followed by further doses based on TDM, are the recommended doses of aminoglycosides [76,77]. For aminoglycosides, a C_max_/MIC ≥ 8–10 is associated with increased efficacy, and toxicity is related to the C_min_, which is taken from a blood sample 30 min or just before the next dosing. The recommended C_min_ for gentamicin and tobramycin is <0.5 mg/L and <2.5 mg/L for amikacin [28].

## 7. Recommendations

TDM is crucial in this patient population, but there is a lack of studies to make recommendations. While waiting for more robust studies, we propose to follow the TDM recommendations and targets of the Infection Section of the European Society of Intensive Care Medicine (ESICM) summarized in Table 1 concerning critically ill patients but not specific to obese patients.

## 8. Conclusions

While TDM is recommended for several antibiotics, there is a lack of data in the literature regarding TDM of antimicrobials in critically ill obese patients, and most studies have only focused on patients with moderate obesity. Therefore, more robust studies should be conducted specifically in the population of obese critically ill patients to better demonstrate the potential advantages of performing routine TDM in this patient population.

## Figures and Tables

**Figure 1 antibiotics-12-01099-f001:**
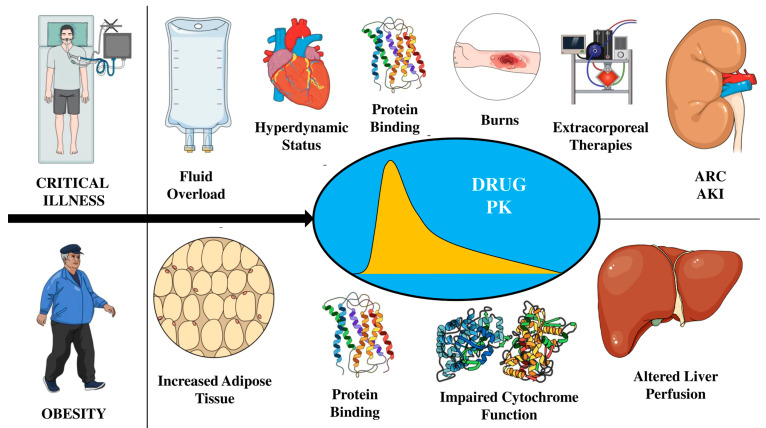
Summary of factors influencing antimicrobial pharmacokinetics (PK) during either critical illness or obesity. ARC = augmented renal clearance; AKI = acute kidney injury.

**Table 1 antibiotics-12-01099-t001:** PK/PD and TDM targets for different antimicrobial agents.

Antimicrobial Class	PK/PD Target	TDM Target
Antifungal Agents		
Azoles		
Voriconazole	C_min_ ≥ 1–2 mg/L	C_min_: 2–6 mg/L (prophylaxis or treatment)
Itraconazole	C_min_ ≥ 0.25–0.5 mg/L (Prophylaxis) C_min_ ≥ 1 mg/L (treatment)	C_min_ > 0.5–1 mg/L
Posaconazole	C_min_ > 0.5 (prophylaxis) C_min_ > 1 mg/L (treatment)	C_min_ > 0.5–0.7 mg/L (prophylaxis) C_min_ > 1 mg/L (treatment)
Fluconazole	AUC_0–24_/MIC ≥ 55–100	/
Antiviral agents	Unclear	/
Antibacterial agents		
β-lactams	50–100% fT > MIC	II: 100% fT > MIC CI: Css > MIC
Glycopeptides		
Vancomycin	AUC_0–24_/MIC ≥ 400 C_min_ > 10–20 mg/L	II: C_min_ ≥ 15–20 mg/L (for severe infections) CI: Css 20–25 mg/L
Teicoplanin	C_min_ ≥ 10 mg/L	C_min_ ≥ 15–30 mg/L
Linezolid	AUC_0–24_/MIC ≥ 80–120 ≥85% fT > MIC	C_min_: 2–7 mg/L
Quinolones	AUC_0–24_/MIC ≥ 125–250 C_max_/MIC ≥ 12	C_max_/MIC ≥ 8–12
Aminoglycosides		
Amikacin	C_max_/MIC ≥ 8–10	C_max_/MIC ≥ 8–10 C_min_ < 2.5 mg/L
Gentamicin	AUC_0–24_/MIC ≥ 110 C_max_/MIC ≥ 8–10	C_max_/MIC ≥ 8–10 C_min_ < 0.5 mg/L
Tobramycin	AUC_0–24_/MIC ≥ 110 C_max_/MIC ≥ 8–10	C_max_/MIC ≥ 8–10 C_min_ < 0.5 mg/L

C_min_ = Trough concentration; MIC = minimal inhibition concentration; II = intermittent infusion; CI = continuous infusion; Css = concentration at steady state; AUC = area under the curve; C_max_ = peak concentration; fT > MIC = time the unbound fraction of the drug remains above the MIC.

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
