# Peer review of "Therapeutic Drug Monitoring of Antimicrobials in Critically Ill Obese Patients"

_antibiotics, 2023, doi:10.3390/antibiotics12071099_

Round 1

Reviewer 1 Report

This is an interesting review, about TDM in obesity, I have just a few comments. 

I don't see methods or search strategy. 

Table 1 - Please add references to the proposed values and targets. 

Please indicate which body weight estimations are used to use in these targets i.e., IBW, ABW etc or at least a short section on how these targets could be interpreted in an obese population.

Author Response

Thank you for the comments.

  1. A short section has been added in the manuscript.
  2. The reference has been added in the manuscript.

the Infection Section of European Society of Intensive Care Medicine (ESICM), Pharmacokinetic/pharmacodynamic and Critically Ill Patient Study Groups of European Society of Clinical Microbiology and Infectious Diseases (ESCMID), Infectious Diseases Group of International Association of Therapeutic Drug Monitoring and Clinical Toxicology (IATDMCT), Infections in the ICU and Sepsis Working Group of International Society of Antimicrobial Chemotherapy (ISAC), Abdul-Aziz MH, Alffenaar JWC, et al. Antimicrobial therapeutic drug monitoring in critically ill adult patients: a Position Paper#. Intensive Care Med. juin 2020;46(6):1127‑53.

Reviewer 2 Report

Reviewer comments

This manuscript provided a concise review of the status of therapeutic drug monitoring (TDM) of antibiotics in critically ill obese patients. The contents of the review fit the scope of the journal. Although the manuscript is well organized, this review lacks conclusive guidance on the therapeutic drug monitoring (TDM) of antibiotics in this patient population.

I have a few comments, both major and minor, to further improve the quality of the manuscript.

Major:

1.     Page 2 “The AUC/MIC ratio may also be affected, as AUC is a function of both CL and Vd.”

This statement looks problematic, the AUC at steady state should only be affected by CL (without Vd) if linear PK, given the equation AUCss = F*DOSE/CL

2.     Please adjust the title of the manuscript to include the word “antibiotics” as this review is very specific to antibiotics use in this patient population.

3.     Overall, this review lacks conclusive recommendations for antibiotic uses in obese critically ill patients. Totally understand if not many studies being conducted in this field. If this is the case, more discussion on the overall significance of this review should be added in the conclusion section.

Minor:

1.     Page 2 “…while lipophilic antimicrobials are cleared hepatically”. Please add word “primarily” somewhere

2.     Page 2 “Critically ill patients may experience both extremes of renal clearance, including acute renal failure (ARF) and augmented renal clearance (ARC). In the case of multiple organ dysfunction syndrome, including renal and/or hepatic dysfunction, antimicrobial clearance may be reduced, potentially leading to toxicity due to antimicrobial accumulation.”

Please insert references.

3.     Page 2, “Therefore, hypoalbuminemia is a factor that contributes to the greater Vd and CL of many antimicrobials [17,18] in critically ill”, lacks a word “patients” at the end

4.     Page 3, not sure what is this “Table 1. This is a table. Tables should be placed in the main text near to the first time they are cited.”

5.     Page 3, “40 kg/m2”. 2 should be superscripted.

6.     Page 3, “PK/PD changes have been summarized in Figure 1.” Should be “Factors introducing PK/PD changes have been summarized in Figure 1.”

7.     Page 4, “Therapeutic drug monitoring of anti-infectious agents has been used since the 1960s to prevent toxicity, therapeutic failure and the emergence of bacterial resistance [31]. Despite the increasing evidence of TDM's utility, its routine use is limited due to cost, availability [32,33], long turnaround time to obtain results in many laboratories (except for aminoglycosides and glycopeptides), and limited data showing favorable clinical outcomes when TDM is performed.”

The position for references looks weird in this sentence, looks like only bacterial resistance and cost, availability have associated references. If so, please provide relevant references to all the other factors listed here.

8.     Page 4-7, please rearrange the subtitle hierarchical structure.

For example, it looks like “Antiviral agents” should be 5.2, “Antibacterial drugs” should be 5.3, etc

9.     Table 2, table notes:

“Cmin= trough” should be “Cmin= trough concentration”

“CI= cotinuous infusion;” should be “CI= continuous infusion;”

“Cmax= peak” should be “Cmax= peak concentration”

Please see minor comments above. 

Author Response

Thank you for all the suggestions.

Major :

  1. Changes have been made in the manuscript.
  2. The title has been modified.
  3. This is our conclusion.

“TDM is crucial in this patient population,  but there is a lack of studies to make recommendations. While waiting for more robust studies, we propose to follow the TDM recommendations and targets of the Infection Section of the European Society of Intensive Care Medicine (ESICM) summarized in Table 1 concerning critically ill patients but not specific to obese patients”.

Minor :

  1. Changes have been made in the manuscript.
  2. This is the reference.

Roberts, J.A.; Abdul-Aziz, M.-H.; Lipman, J.; Mouton, J.W.; Vinks, A.A.; Felton, T.W.; Hope, W.W.; Farkas, A.; Neely, M.N.; Schentag, J.J.; et al. Individualised antibiotic dosing for patients who are critically ill: Challenges and potential solutions. Lancet Infect. Dis. 2014, 14, 498–509. https://doi.org/10.1016/s1473-3099(14)70036-2

3. Changes have been made in the manuscript.

  1. Table 1 has been moved in the text.
  2. Changes have been made in the manuscript.
  3. Changes have been made in the manuscript.
  4. The reference has been moved in the sentence.
  5. The subtitle hierarchical structure has been changed.
  6. Changes have been made in the manuscript.

Round 2

Reviewer 2 Report

The author addressed my previous comments adequately.